

# Physiological changes and gene responses during *Ganoderma lucidum* growth with selenium supplementation

Bo Zhang, Wei Tan, Jie Zhou, Lei Ye, Dinghong Jia and Xiaolin Li

Sichuan Institute of Edible Fungi, Chengdu, Sichuan, China

## ABSTRACT

*Ganoderma lucidum basidiomycota* is highly appreciated for its health and nutrition value. In the present study, *Ganoderma lucidum* was cultivated as selenium transformation carrier, and the physiological changes and gene responses by selenium supplementation were revealed through high-throughput RNA-Seq technology. As a result, selenium supplementation increased the stipe length and the cap size, but decreased the cap thickness of *G. lucidum*. Mineral salt supplementation could greatly promote the formation of triterpene acids and selenium in *G. lucidum*. The highest yield was gained in the treatment with selenium content of 200 μg/g. Subsequently, the tissues of *G. lucidum* at budding and mature stages in this treatment group were sampled for transcriptome analysis and compared to those of a control group without selenium supplementation. A total of 16,113 expressed genes were obtained from the transcriptome of *G. lucidum*, and GO-annotated unigenes were mainly involved in molecular functions and KEGG-annotated ones were highly expressed in ribosomal pathway. Furthermore, genes involved in carbon metabolism pathway were most promoted by selenium at budding stage of *G. lucidum*, while gene expression was the highest in the pathway of amino acid biosynthesis at mature stage of *G. lucidum*. Specially, selenium-related genes in *G. lucidum*, such as GL23172-G, GL29881-G and GL28298-G, played a regulatory role in oxidoreductase, antioxidant activity and tryptophan synthesis. The results provide a theoretical basis for further study of selenium-enriched mushrooms and aid to development of Se-enriched foodstuff and health products made from fungi.

## INTRODUCTION

*G. lucidum* is an edible fungal species in the phylum *Basidiomycota*. Their fruit bodies and spores are abundant in health and nutritious substances including polysaccharides, triterpenoids and nucleoside (*Yu & Zhai, 1979*; *Kinahan, Kowal & Grindey, 1981*). These substances are proved to strengthen immune system and inhibit tumor formation (*Wang et al., 2002*; *Sakamoto et al., 2016*), hence usually utilized in clinical medicine. With great medicinal value, *G. lucidum* becomes appreciated and has been cultivated in large areas in China. Moreover, a long history of *G. lucidum* cultivation contributes to sophisticated planting skills, as well as its high and stable yield (*Boh et al., 2007*). To satisfy the expanding

Corresponding authors
Wei Tan, 332528058@qq.com
Xiaolin Li, kerrylee_tw@sina.com

market demand of *G. lucidum* products, exploration in improving its quality and growth efficiency is now pursued.

Selenium is a multifunctional bionutrient element, and is also recognized as the necessary trace element for important metabolic enzymes. Moreover, as a key component of glutathione peroxidases and selenoproteins, selenium shows great significance to human health with anti-oxidation, anti-tumor, enhancing immunity, *etc*. (*Rayman, 2012*; *Rayman, Infante & Sargent, 2008*; *Rotruck et al., 1973*). The main existence way of selenium in nature is inorganic, and another way is organic in the plant with low concentrations. It is evidently demonstrated that inorganic selenium is more toxic and difficult to absorb compared to the organic state. So it is increasingly demanding to seek a way of transformation. A large number of edible mushrooms (*e.g.*, *Flammulina velutipes*, *Pleurotus ostreatus* and *Ganoderma lucidum*) were reported to be capable of selenium accumulation and transformation, becoming an ideal Se-enriched foodstuff (*De Souza et al., 1999*; *Hanson et al., 2003*; *Haddad et al., 2013*; *White, 2015*; *Zayed, Lytle & Terry, 1998*). Thus, researches have focused on mineral enrichment in edible fungi, with expectation of transforming the supplemented elements from inorganic to organic states. Zhao and Hartman cultivated *G. lucidum* and other mushrooms with essential element addition (*e.g.*, selenium and calcium) in the substrates, and finally harvested nutritional value-improved fruit bodies (*Hartman et al., 2000*; *Zhao et al., 2004*). Our previous study demonstrated the cultivated *Auricularia cornea* with a 100 µg/g supply of selenium in the substrate outperformed with high yield, rich crude polysaccharides and selenium content (*Li et al., 2019*).

Illumina high-throughput technology is widely applied to transcriptome sequencing and exploration of gene composition and functions for mushrooms based on its unprecedented handling capacity, scalability and speed (*Patel & Jain, 2015*). Also, this technology is regarded as a necessary way to clarify the biosynthetic pathways of bioactive compounds that mushrooms produce (*Tomohiro, 2021*). *Dong et al. (2021)* identified 17 candidate genes that were involved in triterpenoid biosynthesis using high-throughput sequencing technology, getting a molecular understanding of *Phellinus igniarius*. *Duan, Bao & Bau (2021)* performed high-throughput transcriptome sequencing of a wild mushroom species *Leucocalocybe mongolica*, and discovered expression changes of some key CAZyme-related genes between mycelia and fruiting body organs. Additionally, real-time quantitative PCR becomes increasingly important in the quantitative detection of genes for its obvious advantages (*Pfaffl, Horgan & Dempfle, 2002*). qPCR methods have been used in detection of gene stability and verification of gene functions (*Zarivi et al., 2015*; *Li et al., 2019*).

Despite some studies on *G. lucidum* as a transformation carrier of mineral elements like selenium, the suitable concentration and mechanisms of selenium accumulation in *G. lucidum* should be further explored. Physiological changes including mycelial growth rate, stipe length, cap size and thickness, fresh yield, contents of the crude polysaccharide, triterpenoids and total selenium in the mature fruit bodies and gene responses during *G. lucidum* growth by selenium supplementation were investigated in this study using high-throughput sequencing technology. Six differentially expressed genes, which were

potentially selenium-dependent were selected for real-time reverse transcription PCR (RT-PCR) to validate the gene expression profiles in *G. lucidum* transcriptome.

## MATERIALS AND METHODS

### *Ganoderma lucidum* cultivation

The studied *G. lucidum* strain was Chuan Yuanzhi No. 1. The *G. lucidum* cultivar 'Chuanyuanzhi 1' was derived from Fujian Province, China and the species was proved to be *G. lucidum* through systematic breeding in Sichuan. Now the cultivar has been deposited in the China General Microbiological Culture Collection Center (CGMCC) with a strain number CGMCC 13174. The authentication of Chuan Yuanzhi No. 1 was published in Acta Horticulturae Sinica in 2017 (*Chen et al., 2017*). The substrate was composed of 90% cottonseed hull, 5% wheat bran, 4% corn flour and 1% gypsum, and a sodium selenite ($Na_2SeO_3$) solution was supplemented to the substrate. The mixed substrate was put into polypropylene cultivation bags with the size of 17 cm × 33 cm × 0.005 cm. The final concentrations of sodium selenite in the substrate were determined to be 0, 50, 100, 200, 250, and 300 μg/g, with the labels of GCK, G50, G100, G200, G250, G300, respectively. The substrate bags were then under sterilized at 98–100 °C for more than 18 h. Afterwards, the bags were cooled to room temperature and then prepared for inoculation. Inoculation was done in a laminar flow cabinet with a sterile environment, and then moved into a culture room for cultivation at 25–28 °C. With the mycelia full of bags, they were planted in the cultivation site at Zhaojia, Jintang, China (N 30°48′16.45″, Ei 104°35′48.79″). The space of the cultivation site was ventilated, and previously cleaned and simply disinfected with lime.

### Growth index investigation

The mycelial growth rate of *G. lucidum* was measured before the mycelia were full of cultivation bags. The mycelium growing edge on the 7th and 14th day after mycelium germination were marked, respectively. Then a ruler was used to investigate the length, and mycelial growth rate per day was calculated. Other growth indexes including stipe length, cap size and thickness, fresh yield, crude polysaccharide, triterpenoids and total selenium concentration were determinated at the mature stage. The cap of *G. lucidum* was regard as a oval, and the long diameter (a) and short diameter (b) were measured, and the cap size (S) could be calculated following the formula: $S = (\pi ab)/4$. While stipe length and the thickness were directly measured with a ruler and a vernier caliper, respectively. The fresh fruit bodies of different bathes in the same treatments were harvested and weighed. The yield was calculated with total weight divided by the number of cultivated bags. The principle of the method used for crude polysaccharide determination was based on "Determination of crude mushroom polysaccharides" of Agricultural Industry Standards of the People's Republic of China (*Xing et al., 2008*) while the triterpenoids was determined with a TU-1810 ultraviolet spectrophotometer (*Yan et al., 2017*). The total selenium content was determined by fluorescence method with an atomic fluorescence meter according to the National Food Safety Standard (GB 500993-2010). The statistical analysis was done using Excel and SPSS13.0.

## Sample collection for transcriptome analysis

The *G. lucidum* tissues of GCK and G200 were sampled at both budding and mature stages with prepared gloves, tweezers and knives, which were previously sterilized. There were four samples, labelled GCKb, GCKm, G200b and G200m, respectively. Each sample had three replicates. Firstly, 3–4 complete fruit bodies from the same batch in the same treatment were crushed and uniformly mixed, and then more than 500 mg of fresh tissues per replicate were collected and pooled from the mixed samples. Afterwards, the samples were stored in liquid nitrogen with 2 mL Eppendorf tubes (Eppendorf, Germany), and then sent to Personalbio (Shanghai, China) for RNA extraction and transcriptome sequencing. The statistical power of this experimental design, calculated in R (version 4.1), was 0.5696 in RNASeqPower.

## RNA extraction, library preparation and sequencing

Total RNA of *G. lucidum* samples was extracted with a Qiagen RNeasy mini kit (Qiagen, Germany) according to the manufacturer's instructions. After assessing RNA quality and contaminated RNA elimination, the remaining RNA was cleaved into fragments of 200–300 bp in length. Then the RNA was reverse transcribed to cDNA with a Fast Quant RT Kit (TIANGEN, Beijing, China). With libraries diluted to 2 nM uniformly and formed single stranded with alkaline denaturation, they were finally paired-end sequenced based on Illumina NextSeq 500 (Illumina, San Diego, CA, USA). All the raw sequences were deposited in the NCBI Sequence Read Archive (SRA) database with the accession No. SRR5576791–SRR5576802.

## Assembly and annotation

Raw reads with FASTQ format were checked and filtered using the FastQC program (*Lu, Tzovaras & Gough, 2021*). The adapters were removed, and then sequences shorter than 50 bp or lower than Q20 in quality score were removed. Afterwards, *de novo* assembly was performed with the program Trinity (*Zhou et al., 2014*; *Grabherr et al., 2011*). A total of $3.72 \times 10^8$ raw reads were obtained in the study, with approximately 0.99% of low-quality reads removed, and finally $3.69 \times 10^8$ high-quality reads were screened out. Besides, the statistical results of Q20, Q30 and GC contents related to the obtained sequences were in Table S1. Furthermore, each sample contained 4.45 GB data, and 67.43% of high-quality sequences were aligned to the corresponding reference genome, so as to carry in-depth analysis of transcriptome data.

    The longest sequence in each cluster was treated as one unigene, and annotated against the databases of GO (Gene Ontology) (*Ashburner et al., 2000*), KEGG (Kyoto Encyclopedia of Genes and Genome) (*Kanehisa et al., 2000*), KOG (Cluster of eukaryotic Orthologous Groups), NR (Non-Redundant Protein Sequence Database) and SwissProt (Swiss-Prot protein) (*Boeckmann, 2003*). The clean sequences were aligned to analyze differential gene expression and enrichment. We used KOBAS software to perform KEGG pathways enrichment analysis following the hypergeometric distribution principle (*Gentleman et al., 2009*).

**Table 1 *G. lucidum* agronomic traits and nutrient content of fruiting bodies in different treatments.**

| NO. | MGR (mm/d) | Length (cm) | Size (cm²) | Thickness (cm) | Yield (g/bag) | CP (%) | TT(%) | TSC (µg/g) | AR (%) |
|---|---|---|---|---|---|---|---|---|---|
| GCK | 5.26 ± 0.37 c | 3.13 ± 0.70 a | 35.53 ± 10.20 bc | 2.35 ± 0.26 a | 125.82 ± 5.56 ab | 0.49 ± 0.08 a | 0.72 ± 0.06 b | 0.18 ± 0.03 e | – |
| G50 | 5.44 ± 0.27 bc | 3.73 ± 0.59 a | 28.94 ± 8.17 c | 1.75 ± 0.14 c | 116.77 ± 3.32 b | 0.60 ± 0.09 a | 0.91 ± 0.01 a | 5.19 ± 0.41 d | 0.90 |
| G100 | 6.38 ± 0.46 a | 3.28 ± 0.77 a | 39.97 ± 10.34 abc | 2.07 ± 0.13 b | 83.01 ± 7.32 c | 0.41 ± 0.05 a | 0.67 ± 0.02 b | 7.48 ± 0.07 c | 0.46 |
| G200 | 5.75 ± 0.42 b | 3.46 ± 0.74 a | 39.43 ± 6.38 abc | 2.13 ± 0.30 ab | 149.50 ± 6.97 a | 0.52 ± 0.24 a | 0.86 ± 0.02 a | 7.59 ± 0.07 c | 0.42 |
| G250 | 5.54 ± 0.65 bc | 3.17 ± 0.45 a | 48.94 ± 15.78 a | 2.02 ± 0.26 b | 91.64 ± 12.10 c | 0.42 ± 0.07 a | 0.88 ± 0.01 a | 11.79 ± 0.47 a | 0.32 |
| G300 | 5.37 ± 0.10 bc | 3.10 ± 1.16 a | 44.61 ± 14.78 ab | 2.17 ± 0.28 ab | 125.48 ± 5.84 ab | 0.61 ± 0.07 a | 0.91 ± 0.01 a | 9.73 ± 1.06 b | 0.30 |

Note:
NO, cultivating formula; MGR, mycelial growth rate; Length the length of a single random *G. lucidum* stipe; Size, the size of a single random *G. lucidum* cap; Thickness, the thickness of a single random *G. lucidum* cap; Yield, the fresh yield per bag; CP, the content of the crude polysaccharide in the mature fruit bodies; TT, the content of the triterpenoids in the mature fruit bodies; TSC, the total selenium concentration in the mature fruit bodies; AR, accumulation rate of sodium selenite, $AR = (TSC \times Dry\ yield)/(Sodium\ selenite\ concentration \times Dry\ substrate\ weight)$, Dry yield = Yield × 0.38, Dry substrate weight = 0.51 kg. GCK, the control group without the addition of selenium; G50, the treatment group with 50 µg/g sodium selenite addition; G100, the treatment group with 100 µg/g sodium selenite addition; G200, the treatment group with 200 µg/g sodium selenite addition; G250, the treatment group with 250 µg/g sodium selenite addition; G300, the treatment group with 300 µg/g sodium selenite addition. Data with different lower-case letters display significant differences ($p$-value < 0.05) by the LSD method using a one-way ANOVA. MGR, Length, Size, Thickness and Yield were replicated 8–10 times, while CP, TT and TSC were replicated three times.

## Gene expression analysis and validation

RSEM was used for expression quantification of RNA-Seq data with a reference of *de novo* assembled transcriptome, and the results of gene alignment was investigated (*Li & Dewey, 2011*). Based on RNA-Seq technology, each unigene's FPKM value that represented the expected fragment numbe per kilobase of transcript sequence per million sequenced reads (*Trapnell et al., 2010*) was calculated as the expression level. Analysis of unigene expression difference was carried out with DESeq (Version 1.18.0) (*Li & Dewey, 2011*). The expressed genes with significant difference (DEGs) were screened and the threshold for screening was |log2(FoldChange)| > 1 and *p*-value < 0.05. Furthermore, a heatmap was drawn to display the expression pattern of each DEG across all the samples between the selenium-treated (G200) and control groups (GCK) with two-way hierarchical clustering based on the R package Pheatmap (*Tauno & Jaak, 2015*). Four identified unigenes were selected for expression validation using the qPCR analysis. A Super RT Kit (TaKaRa, Osaka, Japan) was for RNA reverse transcribing, and Ribosomal Protein L4 was used as reference to amplify and normalize gene expression for each qPCR using primers (*Xu et al., 2014*). Ultimately, each gene expression in one sample was confirmed with not less than three independent qPCR reactions.

## RESULTS

### *G. lucidum* growth changes affected by selenium supplementation

Selenium supplementation significantly affected the physiological development of *G. lucidum* including agronomic traits and nutrient contents in the study (Table 1, Fig. 1). The mycelial growth was evidently promoted by selenium supplementation, and the treatment with the selenium concentration of 100 µg/g in the substrate yielded the fastest (6.38 mm/d), significantly faster than the control (P < 0.05). Selenium supplementation showed a limited effect on *G. lucidum* shape characteristics, including the stipe length and cap size of fruit bodies, had no significant changes in each treatment. While the cap thickness of Se-treated fruit bodies was smaller than that of the control, those of G50, G100

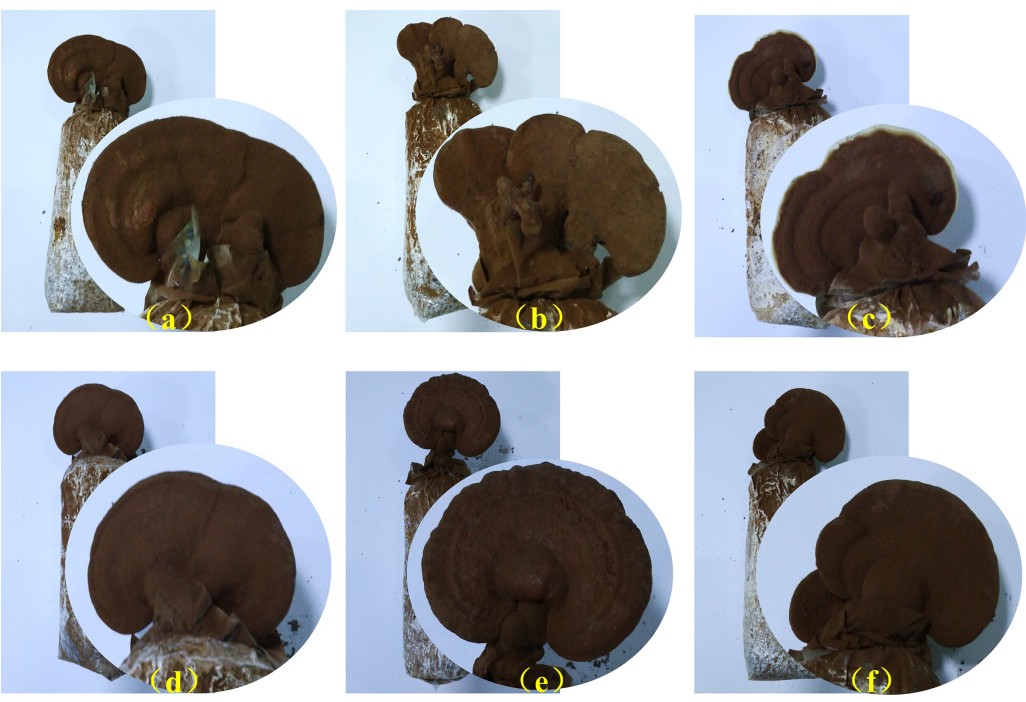

**Figure 1 Fruit bodies of *G. lucidum* at the mature stage in different treatments.** (A) the control group without the addition of selenium; (B) the treatment group with 50 µg/g sodium selenite addition; (C) the treatment group with 100 µg/g sodium selenite addition; (D) the treatment group with 200 µg/g sodium selenite addition; (E) the treatment group with 250 µg/g sodium selenite addition; (F) the treatment group with 300 µg/g sodium selenite addition.

and G250 were significantly thinner than the control. G200 had the highest fresh yield (149.50 g/bag), which was 12.44% higher than the control, while the other treatments delivered lower yields. In addition, selenium supplementation had different effects on the active components of fruit bodies Triterpenoid acids in the *G. lucidum* fruiting bodies were promoted, and the contents increased with selenium supplementation from 100 to 300 µg/g. The highest triterpenoid acids (0.91%) were found in the treatments with the selenium concentration of 50 and 300 µg/g in the substrate, significantly higher than the control. However, there was no significant difference between the polysaccharide contents in fruit bodies of each treatment, and G300 was the highest, reaching 0.61%.

The total selenium in the fruit bodies increased first and then decreased with the selenium addition. The Se-treated fruit bodies were significantly higher than the control in selenium accumulation. It peaked at 11.79 µg/g in the treatment with the selenium concentration of 250 µg/g in the substrate. Treatment with the lowest selenium concentration (G50) had the highest accumulation rate of sodium selenite (0.90%), and then it decreased by selenium supplementation. Thus, it demonstrated that *G. lucidum* is capable of selenium absorption but with a cumulative threshold. Consequently, *G. lucidum* cultured in the substrate with 200 µg/g of selenium was selected for transcriptome analysis to explore gene responses by selenium supplementation at the budding and mature stages due to its highest yield and rich nutrients.

**Table 2 Gene number investigation in KEGG database at budding and mature stages.**

| KEGG | GCKb Up number | G200b Up number | Gb Total number | GCKm Up number | G200m Up number | Gm Total number |
|---|---|---|---|---|---|---|
| Cellular processes | 25 | 62 | 576 | 42 | 40 | 573 |
| Environmental information processing | 19 | 102 | 437 | 54 | 27 | 461 |
| Genetic information processing | 15 | 76 | 815 | 30 | 58 | 826 |
| Metabolism | 246 | 419 | 1,800 | 230 | 191 | 1,766 |
| Organismal systems | 23 | 138 | 657 | 98 | 32 | 687 |

Note:
GCKb, the control group without the addition of selenium at budding stage; G200b, the treatment group with 200 µg/g sodium selenite addition at budding stage; GCKm, the control group without the addition of selenium at mature stage; G200m, the treatment group with 200 µg/g sodium selenite addition at mature stage.

## Functional annotation based on *G. lucidum* transcriptome

The gene expression profiles of *G. lucidum* between the selenium-treated samples at 200 µg/g concentration (G200) and the control (GCK) at the budding and mature stages were analyzed using High-throughput RNA sequencing. A total of 16,113 unigenes were obtained, ranging from 50 to 16,000 bp in length. To uncover more *G. lucidum*-related functional genes, five frequently used databases were employed for annotation including GO, KEGG, KOG, NR and SwissProt. As a result, the numbers of the annotated unigenes were 5,554 (GO), 3,139 (KEGG), 5,760 (KOG),10,874 (NR) and 6,328 (SwissProt, Hinxton, Cambridge, UK), accounting for 34.47%, 19.48%, 35.75%, 67.49%, 39.27%, respectively.

The unigenes in the *G. lucidum* transcriptome were totally annotated to 910 GO-terms during its growth period. And 58 of them were with more than 10% of the annotated genes, in which, the category of molecular function had the most terms (29), followed by the biological process (22) and cellular component (seven) (Table S2). There were 22 GO-terms with over 1,000 annotated genes, including binding, catalytic activity, metabolic process, cellular process and so on. Thus, the GO-annotation results indicated GO pathways were regulated by a large number of genes, playing important roles in primordia formation and fruit body maturation of *G. lucidum*.

To further understand the biological pathways during *G. lucidum* growth, the unigenes were mapped to the reference pathways in the KEGG database including cellular process, environmental information processing, genetic information processing, metabolism and organismal systems (Table 2). A total of 4,285 and 4,313 unigenes were annotated at budding and mature stages, respectively. Unigenes related to metabolism were the most abundant at both growth stages, accounting for 42.01% and 40.95%. Results showed that more unigenes in *G. lucidum* were up-regulated at budding stage with 200 µg/g sodium selenite addition in the substrate of all the five KEGG pathways. However, just the pathway of genetic information processing had more up-regulated unigenes in Se-treated *G. lucidum* at the mature stage. Thus, selenium supplementation potentially activated physiological development during *G. lucidum* primordium formation more than at the mature stage.

Three of the KEGG pathways had more than 10 up-regulated genes in the control group at budding stage including biosynthesis of amino acids, carbon metabolism and glycine,

serine and threonine metabolism (Table S3). A total of 19 KEGG pathways had over 10 up-regulated genes with selenite supplemented in the substrate at budding stage, in which, "carbon metabolism" had the largest amount (32), followed by the pathways of "biosynthesis of amino acids" and "citrate cycle (TCA cycle)". However, a smaller amount of unigenes were up-regulated at the mature stage, and just three of the pathways were with more than 10 up-regulated genes in each treatment, representing "carbon metabolism", "peroxisome" and "amino sugar and nucleotide sugar metabolism" in the control group, and "biosynthesis of amino acids", "purine metabolism" and "pyrimidine metabolism" in the Se-treated samples. These database annotations provided basic biological information in *G. lucidium*, contributing to a better understanding of selenium accumulation in the studied fungus.

## Differentially-expressed gene analysis

To generally uncover how selenium addition and growth stage affected the gene expression pattern of *G. lucidum*, a heatmap based on the RPKM method was generated. The growth stage had significant impacts on gene expression, separating the *G. lucidum* transcriptomes into two clusters (Fig. 2). The interesting result was that the up-expressed genes at the budding stage of both treatments became down-expressed at the mature stage. More specifically, the control group had 1,695 up-expressed genes at mature stage and 1,402 up-expressed genes at budding stage (Fig. S1). However, only 975 and 1,291 genes were up-expressed at mature and budding stage with selenium addition, respectively. Also, the gene expression pattern of *G. lucidum* was greatly changed by selenium addition. There were 3,280 differentially expressed unigenes between the selenium-treated and control groups at the budding stage, of which, 1,612 upregulated and 1,668 downregulated in response to selenium addition. In total 912 upregulated and 1,549 downregulated genes were detected with selenium supplementation at the mature stage.

The top 10 up-regulated genes in both treatments (Table S4) and genes with the highest expression (Table S5) at budding and mature stages were investigated in this study. The most up-regulated genes treated with selenium addition were GL23959-G and GL26604-G at budding and mature stages, respectively. The gene GL23959-G was related to the monomeric metabolic process, while the gene GL26604-G was involved in the activity of cation transmembrane transporter. The genes in the Se-treated *G. lucidum* with the highest expression at budding stage was GL21838-G, with relevance to the binding of organic ring compounds, peroxidase activity and stress response, and GL23307-G was the highest at mature stage. Just two up-regulated genes were among the top 10 genes with the highest expression. They were GL23263-G and GL24771-G, annotated in GO:0003824, *etc*.

To validate the gene expression profiles in *G. lucidum* transcriptome, six differentially expressed genes were selected for qPCR. Two of the selected genes GL23172-G and GL29881-G were up-regulated at budding and mature stages with selenium supplementation, and they played a role of regulating the oxido-reductase and antioxidant activities. Meanwhile, the gene GL23172-G, which was related to active oxygen metabolism, was found up-expressed at both growth stages of *G. lucidum*. Moreover, the genes GL24625-G and GL28298-G were both involved in tryptophan synthesis, and one

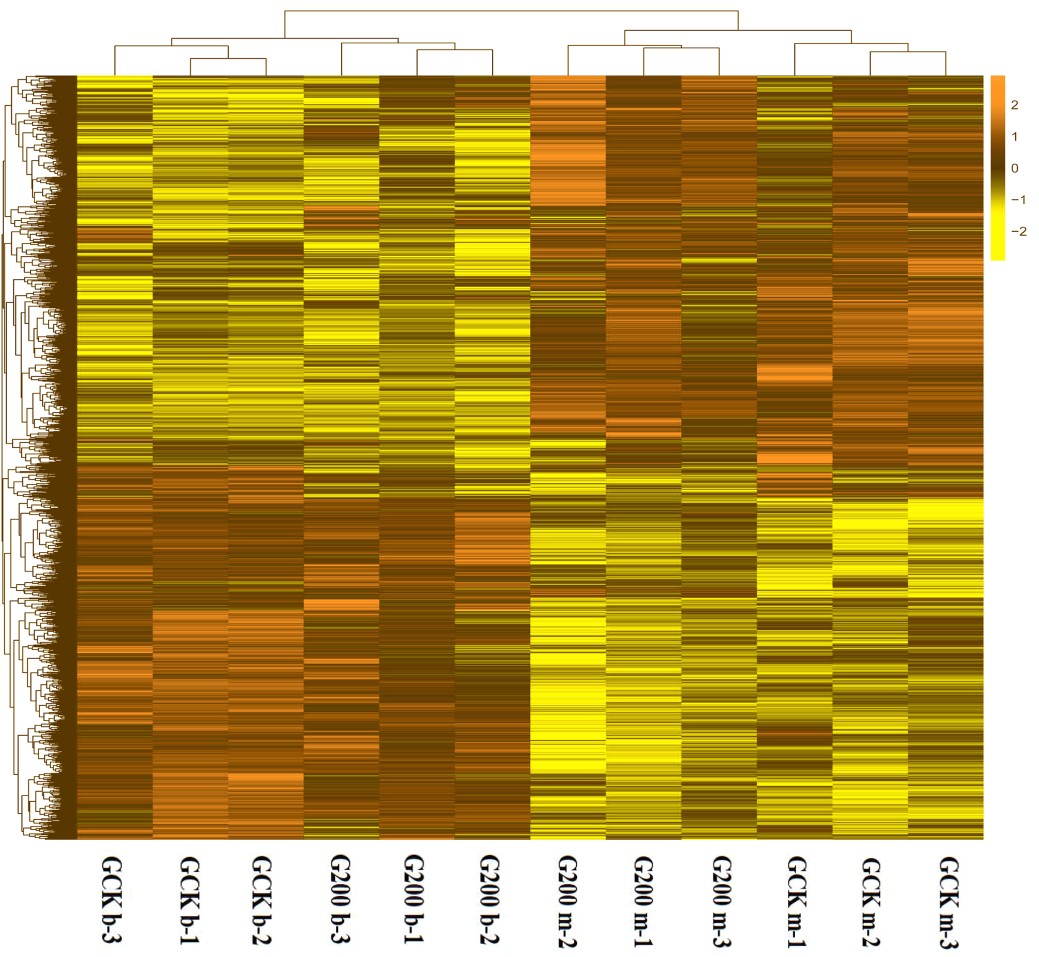

**Figure 2 A heatmap showing the log2 (FoldChange) values of the selenium-responsive DEGs (*n* = 3) in *G. ludium* samples.** The DEGs and samples were subject to bidirectional clustering analysis using the R package Pheatmap based on the Euclidean distance and complete linkage clustering. The up-expressed DEGs are coloured in dark-yellow and the down-regulated DEGs in light yellow, respectively.

was down-regulated and another up-regulated during *G. lucidum* primordium formation. All of the six genes were successfully amplified, and the qPCR results were highly consistent with the DEG expression patterns of the transcriptome analysis, confirming the reliability of the RNA-Seq data (Table 3).

## DISCUSSION

### Selenium supplementation affected *G. lucidum* agronomic traits

One key indicator of hyphal vitality is the growth rate, which also shows the hyphal adaptability to the surroundings. The present study revealed a faster growth rate of *G. lucidum* hypha with selenium supplemented in the substrate, demonstrating that selenium be capable of promoting *G. lucidum* growth at hyphal stage. However, the promotion of growth peaked at the concentration of 100 µg/g. As is reported by *Goyal, Kalia & Sodhi (2015)*, the growth of *G. lucidum* hypha is affected by selenium

**Table 3 Validation of six differentially expressed genes in *G. lucidum* transcriptome.**

| Group | Gene ID | DESeq analysis based on RNA-seq | | Validation of the DEGs by qPCR analysis | | |
|---|---|---|---|---|---|---|
| | | Log2Fold change | *p*-value | GCK ($2^{-\Delta\Delta Ct}$) | G200 ($2^{-\Delta\Delta Ct}$) | *p*-value |
| G200b *vs.* GCKb | GL29881-G | 4.33 | 1.30E−16 | 1.00 ± 0.00 | 1.18E+04 ± 0.00 | 3.09E−12 |
| | GL23172-G | 3.02 | 5.27E−07 | 1.03 ± 0.09 | 9.11 ± 0.00 | 3.38E−05 |
| | GL24625-G | −1.82 | 1.61E−4 | 1.00 ± 0.00 | 4.31E−04 ± 0.00 | 2.67E−04 |
| | GL28298-G | 2.80 | 6.68E−09 | 1.00 ± 0.01 | 10.09 ± 0.00 | 2.30E−06 |
| G200m *vs.* GCKm | GL23172-G | −1.47 | 2.55E−02 | 1.01 ± 0.02 | 0.40 ± 0.00 | 2.79E−02 |
| | GL29881-G | −3.36 | 3.37E−06 | 1.00 ± 0.00 | 0.32 ± 0.01 | 6.70E−03 |

**Note:**
$2^{-\Delta\Delta Ct}$ displays relative gene expression level using RPL4 as a reference genein qPCR analysis. Data are presented as means ± standard deviation of three replicates. Abbreviations: GCK, control group without selenium addition in the substrate; G200, treatment group with 200 μg/g of selenium addition in the substrate; m, at the mature stage; b, at the budding stage.

supplementation, making the hypha thinner and their branches wider. Even the spore morphology is changed by the increase of selenium addition. It is thus clear that higher concentration of inorganic selenium may not be good for hyphal growth. In addition, selenium is found to be widely used in mushroom cultivation, and it can improve the agronomic characteristics of edible fungi (*Wu et al., 2015*). In the present study, selenium supplementation obviously made the cap of *G. lucidum* fruit bodies thinner, while the mineral had no significant effects on the stipe length and cap size. The yield of *G. lucidum* is the most important production indicator that farmers are concerned. The fresh weight of *G. lucidum* fruit bodies was the highest when the selenium concentration was 200 μg/g in the present study. Considering the better agronomic characteristics of the 200 μg/g Se-treated *G. lucidum*, this supplementation concentration can be optimal for *G. lucidum* cultivation. Thus, selenium has a comprehensive effect on *G. lucidum* growth and development, and it still needs further exploration into the unknown mechanism.

## Nutrient contents in *G. lucidum* changed with selenium supplementation

The present study uncovered a continual increase of selenium in *G. lucidum* fruit bodies with selenium supplementation. As is known, the majority of edible fungi (*e.g.*, *G. lucidum*) have a strong ability to enrich mineral elements (*Drewnowska & Falandysz, 2015*; *Kalač & Svoboda, 2000*), and selenium is a typical representative (*Gąsecka, Siwulski & Mleczek, 2018*; *Reilly, 1998*). Besides, selenium is proved to be an indispensable trace element for humans, playing an important role in our health. A dynamic change of selenium enrichment in *G. lucidum* fruit body was revealed, and the largest content was detected at the concentration of 250 μg/g in the present study. According to *Goyal, Kalia & Sodhi (2015)* and *Janssen (2006)*, the selenium absorption by *G. lucidum* hypha shows an increasing trend with the increase of selenium supplementation, which will probably have a direct effect on the selenium accumulation in fruit bodies. The effective adsorption of selenium by *G. lucidum* in our study is consistent with the study of *Li, Guo & Li (2003)* on selenium enrichment of algae. Additionally, it was found that sodium selenite ($Na_2SeO_3$) supplemented into the substrate can be transformed into organic state with bio-absorbing

by edible fungi including *Agaricus bisporus*, *Lentinus edodes* and other mushrooms (*Dernovics, Stefánka & Fodor, 2002*; *Elteren, Woroniecka & Kroon, 1998*; *Ogra et al., 2004*; *Racz et al., 2000*; *Yoshida et al., 2005*). These organic selenium states include selenoproteins and polysaccharides. Also, selenium has been proved to form conjugated complexes with mushroom polysaccharides, significantly improving the biological activities of *G. lucidum* and other edible fungi, such as anti-tumor and free radical scavenging (*Shi et al., 2010*). It follows that *G. lucidum* enriches selenium, aiming to improve its medicinal value. Furthermore, the selenium state in the studied *G. lucidum* fruit bodies must be further verified when they are for sale. As is reported, the suitable selenium intake of human body ranges from 39 to 90 μg per day (*Duffield et al., 1999*). That is to say, only 7.91 g dry fruit body per day can satisfy the selenium need if the studied *G. lucidum* of the highest yield is consumed with organic safety. The selenium supplementation also greatly promoted the triterpene acid contents in *G. lucidum* fruit bodies in the present study, contributing to its higher medicinal properties. In general, selenium can promote the formation of active substances in *G. lucidum* fruit bodies, and the results provide a theoretical basis for the selenium-enriched cultivation of *G. lucidum*.

## Genes from *G. lucidum* transcription responded to selenium supplementation

In this study the obtained 16,113 genes and annotated them into different databases including GO and KEGG, from which the potential genes relating to selenium enrichment were mined. Consequently, catalytic activity is an important functional type during the growth of *G. lucidum*. Different enzymes (*e.g.*, cellulose and lignin peroxidase) participate in the catalytic process during the growth of edible fungi, and biological processes like metabolism, nutrition and energy conversion are greatly determined by catalysis (*Buswell et al., 1996*; *Lechner & Papinutti, 2006*; *Lee et al., 2004*). The gene regulation of *G. lucidum* at the transcriptional level presented significant changes with selenium supplementation in this study. This supplemented element is reported to have antioxidant effect, protecting cells from free radical oxidation damages (*Serafin Muñoz et al., 2006*). Besides, selenium is a cofactor of selenium-related enzymes (*e.g.*, glutathione peroxidase) (*Malinowska et al., 2009*). It was demonstrated by *Goyal, Kalia & Sodhi (2015)* that there are selenium signals present in *G. lucidum* hypha, and they are existing mainly in selenoprotein state (*Janssen, 2006*).

## Selenium-related genes played different roles

Some genes have different functions at different growth stages of organism (*Hsu et al., 2011*; *Muraguchi & Kamada, 2000*). *Yu et al. (2012)* proved proof for this view when they studied the changes in gene expression and biological pathways from the mycelia to the initial primordial stages. It was previously reported that a large number of genes involved in the pathway of bio-synthetic metabolism were up-regulated, while the genes relating to degradation activity presented up-expressed at the stage of fruit formation. Moreover, there was a higher expression of genes coding for hydrophobins and lectins investigated at fruiting stage of *Agaricus bisporus* compared to that at undifferentiated hyphal stage by

*Morin et al. (2012)*. Genes involved in stress signals (*e.g.*, MAPK, cAMP) were found up-regulated at fruiting stage when the gene expressions at different growth stages of *Hypsizygus marmoreus* were compared (*Zhang et al., 2015*). Besides, among the six verified genes, GL28298-G participates in the k00600 pathway, which has something to do with carbon accumulation of folic acid. As is seen in https://www.kegg.jp/kegg-bin/show_pathway?map00670/K01934%09%23FFFFFF,red/K00601%09%23FFFFFF,red/K00288%09%23FFFFFF,red/K00602%09%23FFFFFF,red/K00600%09%23FFFFFF,green/, the k00600 pathway is located importantly in the carbon accumulation process of folic acid (2.1.2.1), greatly contributing to the synthesis of 5,10-methylenetetrahydrofuran and tetrahydrofolate. Moreover, two of the verified genes GL24625-G and GL28298-G were involved in tryptophan synthesis. In summary this study revealed different gene expressions and biological pathways of *G. lucidum* in response to selenium supplementation, which aids to further study of the molecular mechanism in edible fungi.

## CONCLUSION

Our study revealed a significant effect of selenium supplementation on the hyphal growth, morphological characteristics, yield and active substances in *G. lucidum*. The most suitable selenium concentration for *G. lucidum* bag cultivation was selected at 200 μg/g based on. The transcription analysis uncovered the different expressions of some significant genes: the ribosome-related genes were most active during the primordium formation, and the genes related to amino acid biosynthesis were up-expressed during the fruit body maturation. More importantly, the expression of genes in different biological pathways was governed by the growth stage and selenium concentration. Some potential selenium dependent genes were unearthed, which played a regulatory role in oxidoreductase, antioxidant activity and tryptophan synthesis. These results provide a theoretical basis for selenium-enriched mushroom cultivation, helping develop foodstuff and health products of Se-enriched fungi.

### Funding

This work was supported by the China Agriculture Research System (CARS-20) and the Sichuan Mushroom Innovation Team (SCCXTD-2022-07). The funders had no role in study design, data collection and analysis, decision to publish, or preparation of the manuscript.

### Grant Disclosures

The following grant information was disclosed by the authors:
China Agriculture Research System (CARS-20).
Sichuan Mushroom Innovation Team: SCCXTD-2022-07.

### Competing Interests

The authors declare that they have no competing interests.

## Author Contributions

- Bo Zhang conceived and designed the experiments, performed the experiments, analyzed the data, prepared figures and/or tables, authored or reviewed drafts of the article, and approved the final draft.
- Wei Tan conceived and designed the experiments, prepared figures and/or tables, and approved the final draft.
- Jie Zhou performed the experiments, prepared figures and/or tables, and approved the final draft.
- Lei Ye performed the experiments, analyzed the data, authored or reviewed drafts of the article, and approved the final draft.
- Dinghong Jia performed the experiments, analyzed the data, authored or reviewed drafts of the article, and approved the final draft.
- Xiaolin Li conceived and designed the experiments, prepared figures and/or tables, authored or reviewed drafts of the article, and approved the final draft.

## Data Availability

The raw data is available at GenBank: SRR5576791–SRR5576802, PRJNA798550.

## Supplemental Information

Supplemental information for this article can be found online at http://dx.doi.org/10.7717/peerj.14488#supplemental-information.

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
