# Peer review of "Physiological changes and gene responses during Ganoderma lucidum growth with selenium supplementation"

_PeerJ, doi:10.7717/peerj.14488_

## Round 0.1 · original submission · Major Revisions

Three experts in this field assessed your manuscript and found the study interesting and within the scope of this journal. There are concerns related to the methodology that needs to be addressed, among others.

Reviewer 1 ·

Basic reporting

1. The English or article is good, and it is clear, unambiguous, and technically correct text.
2. The article includes a good introduction and background.
3. It needs one profound revision and standardized all references.
4. The structure of the article is the acceptable format for all sections.
5. Figures are relevant to the content of the article. Suggest strongly to carefully review the order of the supplementary tables and their titles.
6. Raw data has been made available by the Data Sharing policy of PeerJ journal define

Experimental design

1. The article has original primary research within Aims and Scope of the journal
2. The article clearly defines the research question.
3. The study contributes to filling the gap in the basic physiology of G. lucidum with selenium supplementation
4. The work has a high technical standard.
5. Methods were described with sufficient information to be reproducible by another investigator

Validity of the findings

1. The results are exciting and helpful to understand the metabolic response of G. lucidum to the presence of high concentrations of selenium in the culture medium.
2. The data are robust, statistically significant, and with controls.
3. The conclusions are reasonable in general. Could you go deeper into why the selected genes impact the tryptophan pathway and not the cysteine synthesis? Is there in this fungus selenocysteine amino acid?

Additional comments

no comments

·

Basic reporting

The manuscript suffers from professional English usage. I have made changes on the manuscript (by hand) to guide the authors.
The reason for the research is articulated in the introduction section. The hypotheses were not clearly stated but are evident from the information provided by the authors.
The heading of Table 3 should be changed to reflect the information.
Raw data were shared, and all information was in English.
References were sufficient.

Experimental design

The aims and scope fall within the that of the journal. The experimental design was sufficient, the research question was defined and the methods were sufficiently described and appropriate to answer the question. The authors clearly followed the same approach that was used in a paper where some of them were authors. "Li, X., Yan, L., Li, Q. et al. Transcriptional profiling of Auricularia cornea in selenium accumulation. Sci Rep 9, 5641 (2019). https://doi.org/10.1038/s41598-019-42157-2";

Validity of the findings

The results will help to improve the future production of Ganoderma lucidum, and the data generated are novel for this fungal species. The results are sound and robust. The conclusions brought the manuscript to full circle - it linked with the question posed at the beginning.

Additional comments

This study furthers our understanding of the effect of selenium on the growth and yield of edible mushrooms. Earlier research similar to the work presented in the manuscript was done in the paper "Li, X., Yan, L., Li, Q. et al. Transcriptional profiling of Auricularia cornea in selenium accumulation. Sci Rep 9, 5641 (2019). https://doi.org/10.1038/s41598-019-42157-2"; Thus, the research in the submitted manuscript complements the data generated for a few mushroom-forming edible fungi.

My main concern is with the written part of the manuscript. I have made some suggestions on the manuscript for the authors to consider (my apologies for them being hand-written, I hope it will be readable).

I also request that the authors indicated how they made sure that they were indeed working the Ganoderma lucidum. The reason for my request is due to the fact that G. lucidum is species complex - were the authors really working with G. lucidum s.s.? Furthermore, if this strain was obtained from a culture collection, then one should ensure that the strain that one receives is indeed that species (cultures get mixed up sometimes in collections). Also, more information should be provided about the strain.

Also, please see my comments about the title of Table 3 - the current title does not link with the information presented in the table.

In addition to Figure 1, I suggest that the authors include images showing the growth of the Ganoderma lucidum cultures at the different selenium concentrations.

Reviewer 3 ·

Basic reporting

The manuscript is clear and fairly easily to read. However, language corrections are recommended for some parts of the manuscript. In addition, I would like to suggest the use of correct terminologies such as pileus (for cap) and basidiocarp (for fruit/fruiting bodies).

Experimental design

The experimental design is not clear and there is a lack of details in the methodology section. One major comment is the identity of fungal species used in the present study (labeled as Ganoderma lucidum). There is no mention if the species was authenticated by any mycologist and/or using any molecular methods. In recent years, there is a lot of confusions regarding the proper scientific name for the fungal species previously known as Ganoderma lucidum. Please consider this matter.

Other issues that require clarification are as follow:

Line 79: please explain how mycelial growth rate was measured - on what substrate/media?
Line 80: please explain how cap size and thickness and stipe length was measured
Line 81: please indicate the instrument used to determine the concentration of selenium and the method should be referenced
Line 81: what is/are the parameter(s) to determine "mature stage"
Lines 82-83: please indicate the standards used for the total polysaccharides and triterpenoids assay, please edit the units in Table 1 accordingly
Line 89: the replicates come from the same fruiting body or different fruiting body, same batch or different batch - it will be useful to include photographs of the samples used
Table 1: please explain how the yield was calculated

This section does not meet the criteria of PeerJ.

Validity of the findings

While a lot of data has been collected, there is little coherence in the discussion on the relationships between the physical and biochemical changes in the basidiocarp with increasing level of Se supplementation. In fact, in the discussions section, there is an emphasis on the nutritional benefits of Se and Se-enriched food which are not investigated in the present study, while very little postulations on the relationship between Se supplementation and the observed changes.

The main weaknesses of this study is the manner by which the concentration of Se in the basidiocarp was analysed and determined was not stated. Even with the gene expression data, it is unclear how Se supplementation might have altered the physical parameters and biochemical content. Claims like "... promote the formation of active substances (line 260)" need to be discussed based on previous findings and substantiated with data. Moreover, colorimetric assays were used to estimate the "total triterpenoids" which do not provide any clues on the abundance of different triterpenoids (not all are bioactive). Such assays also suffer potential interferences by other chemicals. Data from chromatography or mass spectrometry will be useful to track the differences in concentration of selected bioactive triterpenoids.

Line 255-258: the discussion on how consuming Ganoderma can fulfill the daily requirement of Se is too premature to be discussed here because the woody and fibrous Ganoderma is not even eaten (directly) as food in the first place. Ganoderma is usually used for medicinal purposes and the study here does not take into account how Se will be affected by the different preparation (extraction) method. Please consider to limit or remove the extensive information on the nutritional benefits of Se and Se-enriched food.

Additional comments

The authors aimed to investigate the effect of Se supplementation on the physiological parameters of a fungal species denoted as Ganoderma lucidum and to compare the gene expression for the treated and non-treated samples. Issues related to the authentication of fungal species and the lack of clarity in the methodology ought be to be given attention. Some differences in pileus and stipe of the basiocarp were observed but the variation is minimal. The authors are suggested to include photographs of the pileus and stipe. The biochemical content should be expressed in the correct unit. It is difficult to explain the effect of Se supplementation on Ganoderma lucidium even with the data from physical, biochemical and gene expression studies (refer above). Careful interpretation of the results, in the context of previous findings, is suggested.

---

## Round 0.2 · Major Revisions

One Reviewer thinks the manuscript is suitable for publication, the other one does not. She/He still thinks the authentication of mushroom material being investigated in this study is crucial. I agree with this comment and also see this aspect as essential for this work.

Reviewer 1 ·

Basic reporting

Basic Reporting

1. The authors have considered and responded to the reviewers' suggestions.
2. The manuscript is straightforward and quick to read.
3. Language corrections were attended

Experimental design

Experimental design

The experimental design is correct. In addition, the methods used are well-defined and adequate to meet the objectives stated in the manuscript.

Validity of the findings

Validity of the findings
The results will help to understand the metabolic processes involved in the presence of Se. Furthermore, It will help to improve the future production of Ganoderma lucidum.
The data resulting from the transcriptomic analysis are novel for this fungal species. However, the results are solid and robust.
The conclusions link to the question posed at the beginning.

Additional comments

Additional comments
The authors aimed to investigate the effect of Se supplementation to enrich the nutraceutical value of the mushroom Ganoderma lucidum and to compare the gene expression of Se-treated and non-treated samples to determine what changes in mushroom physiology occur in the presence of Se.

1. Revise the numbering of the supplementary tables.
2. On line 116 corrected ...X108
3. the references of lines 245, 335, 319, and 358 are incomplete

Reviewer 3 ·

Basic reporting

The manuscript is fairly well-written.

Experimental design

Lines 89-91. Please provide the principle of the methods used . For the total triterpenoid measurement, what was the standard used? The reference Zhang 1987 described methods for polysaccharides but it was cited under triterpenoid analysis. Please check.

Validity of the findings

The authentication of mushroom material being investigated in this study is crucial. While I understand the authors obtained their samples from reliable sources, the authors must consider the developments in the taxonomy of a mushroom previously thought to be Ganoderma lucidum (used extensively in all parts of the world, including China) may not be G. lucidum after all. Some proposed to call it Ganoderma lingzhi, while others proposed other names. Therefore, I strongly feel that molecular identification of the samples used in this study must be undertaken to prevent confusions in the future.

For further reading:
Distinguishing commercially grown Ganoderma lucidum from Ganoderma lingzhi from Europe and East Asia on the basis of morphology, molecular phylogeny, and triterpenic acid profiles
https://www.sciencedirect.com/science/article/pii/S0031942216300413

Cao, Y., Wu, SH. & Dai, YC. Species clarification of the prize medicinal Ganoderma mushroom “Lingzhi”. Fungal Diversity 56, 49–62 (2012). https://doi.org/10.1007/s13225-012-0178-5

Loyd AL, Richter BS, Jusino MA, Truong C, Smith ME, Blanchette RA and Smith JA (2018) Identifying the “Mushroom of Immortality”: Assessing the Ganoderma Species Composition in Commercial Reishi Products. Front. Microbiol. 9:1557. doi: 10.3389/fmicb.2018.01557

Other relevant papers.

Additional comments

The authors have attempted to address all queries raised by the reviewers. Thank you. The authentication of the mushrooms used is very important as similar concern was raised by another reviewer. Please consider the above suggestions for the benefit of the scientific community in the future.

In terms of technicality, please insert a clear, high-resolution image for Figure 1 and to consider to include magnified images of the fruiting bodies.

---

## Round 0.3 · accepted · Accept

The manuscript was significantly improved after addressing the Reviewers' comments and is now suitable for publication in PeerJ.

Reviewer 3 ·

Basic reporting

No comment

Experimental design

No comment

Validity of the findings

No comment

Additional comments

The authors have confirmed that molecular identification of the species used in their study was performed and published earlier, and cited that in the revised manuscript. I do not have further queries.